



# The Importance of Initial Conditions in Seasonal Predictions of Antarctic Sea Ice

Elio Campitelli[1, 2], Ariaan Purich[1, 2], Julie Arblaster[1, 2], Eun-Pa Lim[3], Matthew C. Wheeler[3], and Phillip Reid[3]

[1]School of Earth, Atmosphere and Environment, Monash University, Kulin Nations, Clayton, Victoria, Australia.
[2]ARC Special Research Initiative for Securing Antarctica's Environmental Future, Clayton, Kulin Nations, Victoria, Australia
[3]Research, Bureau of Meteorology, Melbourne, Australia.
**Correspondence:** Elio Campitelli (elio.campitelli@monash.edu)

**Abstract.** Accurate Antarctic sea-ice forecasts are crucial for climate monitoring and operational planning, yet they remain challenging due to model biases and complex ice-ocean-atmosphere interactions. The two versions of the Australian Bureau of Meteorology's ACCESS seasonal forecast system, ACCESS-S1 and ACCESS-S2, use identical model configuration and differ only in their initial conditions; primarily in that ACCESS-S2 does not assimilate

sea-ice observations, whereas ACCESS-S1 does.

This provides a convenient opportunistic experiment to assess the role of initial conditions on Antarctic sea-ice forecasts using more than 20 years of fully coupled simulations with two 9-member ensembles. Our analysis reveals that both systems experience an extended melt season and delayed growth phase compared with observations. This leads to a significant negative sea-ice extent bias, which is corrected only in ACCESS-S1 by the data assimilation

system. The impact of the differing initial conditions on forecast errors varies dramatically by season: summer and autumn initial conditions (January-April) provide predictive skill for up to three months, with February initial conditions being particularly crucial. In contrast, winter forecasts of the two systems are statistically indistinguishable after only two weeks. Regional analysis of forecast skill suggests that this winter predictability barrier is most dramatic over East Antarctica, where even ACCESS-S1 shows negative skill. These findings highlight the critical importance

of comprehensive year-round sampling in predictability studies and suggest that operational sea-ice data assimilation efforts should prioritise the summer-autumn period when initial conditions have maximum impact on forecast skill.

## 1 Introduction

Accurately modelling Antarctic sea ice is essential for understanding processes and improving climate projections to inform adaptation strategies. Accurate seasonal to sub-seasonal forecasts are also crucial for operation contingency

planning in and around the Antarctic continent, including scientific missions, fisheries, and tourism (De Silva et al.,



2020; Wagner et al., 2020). Improvements in modelled sea-ice might also help improve weather forecasts over and away from sea-ice regions (Rinke et al., 2006; Wang et al., 2024; Semmler et al., 2016).

However, progress in Antarctic sea-ice forecasting system has lagged behind Arctic sea-ice forecasts due to model biases, and inherent large variability and complexity (Zampieri et al., 2019; Gao et al., 2024). Dynamical seasonal forecasts of summer Antarctic sea ice have been shown to perform worse than relatively simpler statistical methods (Massonnet et al., 2023) and machine learning approaches (e.g. Dong et al. (2024), Lin et al. (2025)), which also underscores the need for better understanding and physical modelling of sea-ice dynamics, and drivers of its variability.

Good initial conditions are generally required for a good forecast, however, it is not entirely known to what extent accurate sea-ice initial conditions affect the quality of the forecast and at what timescales. Exploring seasonal predictions of Arctic sea ice, Guemas et al. (2016) found that sea-ice initial conditions are important in autumn to predict summer sea ice, but the impact wasn't as dramatic when predicting winter sea ice. Day et al. (2014) also found seasonally-varying differences in the effect of initialisation, noting that accurate Arctic sea-ice thickness leads to improved sea-ice forecasts initialised in July but not when initialised in January.

For the Antarctic, Holland et al. (2013) studied the initial-value predictability of Antarctic sea ice in a perfect model study using the CCSM3 model. They found that sea-ice and ocean initial conditions provide predictive information to forecast sea-ice edge location several months in advance and that some predictability is retained for up to two years thanks to ocean heat content anomalies that are advected eastward. This is in contrast with Marchi et al. (2020), who ran perfect model experiments to argue that uncertainty in the predicted atmospheric state and evolution is the main driver of uncertainty in Antarctic sea-ice extent prediction on seasonal timescales, with sea-ice and ocean initial conditions having lesser importance. More recently, Morioka et al. (2022) studied decadal forecasts of Antarctic sea ice and found that initialising ocean and sea ice improved the correlation between simulated and observed sea-ice concentration evolution in the Amundsen–Bellingshausen Sea. It is hard to compare these studies since they are based on forecasts initialised at different times of the year and different frameworks: Holland et al. (2013) ran 20 ensemble members initialised on the 1st of January of a particular year, Marchi et al. (2020) ran forecasts from the 1st of March and 1st of September, and Morioka et al. (2022) ran forecasts only from the 1st of March. Marchi et al. (2020) also used a coupled ocean–sea-ice model instead of a fully coupled model like Holland et al. (2013) did. Morioka et al. (2022) used observed sea-ice initial conditions and compared with observations, while Marchi et al. (2020) and Holland et al. (2013) were perfect model studies.

In October 2021 the Australian Bureau of Meteorology (BoM) upgraded the Australian Community Climate and Earth System Simulator – Seasonal (ACCESS-S) from version S1 to S2. While the base model remained the same, the change in version was focused on using ocean, sea-ice and land initial conditions generated by the BoM instead of depending on the UK Met Office. Crucially, compared to ACCESS-S1, ACCESS-S2 does not assimilate sea-ice observations, so sea ice is only affected by the ocean and atmospheric data assimilation via the coupled integration.



Since model configuration is identical between ACCESS-S1 and ACCESS-S2, they form a sort of "opportunistic experiment" where the same forecasting model was run over a long period of time with multiple ensemble forecasts initialised throughout the year, with the only difference being the initial conditions. This provides an opportunity to test the effect of sea-ice initial conditions on the forecast of sea-ice concentrations and the climate.

In this study we compare sea-ice hindcasts produced by ACCESS-S1 and ACCESS-S2. We focus on seasonality of errors and biases and the effect of the data assimilation system. This comparison will inform future work with the prediction system as a research tool to better understand the dynamics and variability of the Antarctic sea ice and its impacts on the climate system as well as to explore the potential of using its sea-ice forecasts for decision-making. The work will also serve as a benchmark for future prediction systems to attempt to improve upon.

## 2 Data and methods

### 2.1 ACCESS-S2

ACCESS-S2 (Wedd et al., 2022) is the Bureau of Meteorology's seasonal forecast system which became operational in October 2021, replacing the ACCESS-S1 system (Hudson et al., 2017). The model components of both ACCESS-S2 and ACCESS-S1 are identical with the same numbers of levels and resolution. They consist of the Global Atmosphere 6.0 (GA6) (Williams et al., 2015; Waters et al., 2017), the Unified Model's Global Land 6.0 (Best et al., 2011; Waters et al., 2017), NEMO Global Ocean 5.0 (Gurvan et al., 2013; Megann et al., 2014) and Global Sea Ice 6.0 [CICE; Rae et al. (2015)]. The atmosphere has a N216 horizontal resolution (~60 km in the mid-latitudes) with 85 vertical levels. The land model uses the same horizontal grid as the atmosphere with four soil levels. The ocean component has a nominal horizontal resolution of 1/4° with 75 vertical levels. The sea-ice component, based on CICE version 4.1, has the same resolution as the ocean component and five sea-ice thickness categories as well as an open water category.

Both systems take atmospheric initial conditions derived from ERA-interim (Dee et al., 2011) for their hindcasts. The main difference between the hindcasts of the two systems are the ocean and sea-ice initial conditions. ACCESS-S1's ocean and sea-ice initial conditions come from the Met Office FOAM system, which uses a multivariate, incremental three-dimensional variational (3D-Var), first-guess-at-appropriate-time (FGAT) data assimilation scheme (Waters et al., 2015) and assimilates sea surface temperature (SST), sea surface height (SSH), in situ temperature and salinity profiles, and satellite observations of sea-ice concentration using the EUMETSAT OSISAF product described in the next section. ACCESS-S2, on the other hand, is initialised from ocean conditions generated by the BoM weakly coupled ensemble data assimilation scheme described in Wedd et al. (2022). This scheme uses an optimal interpolation method and assimilates temperature and salinity profiles from EN4 (Good et al., 2013). SSTs are nudged to Reynolds OISSTv2.1 (Reynolds et al., 2007) in areas where SSTs are over 0°C and Sea Surface Salinity is weakly nudged to the World Ocean Atlas 2013 climatology (Zweng et al., 2013).





Of most relevance for this work, sea-ice concentrations are not assimilated in ACCESS-S2. Assimilation cycles are performed daily. The coupled model runs for 24 hours initialised from the previous cycle. Then the restart file fields of the ocean component are used as first guess in the data assimilation cycle and the innovations are used to build the next ocean initial conditions for the following cycle. The atmosphere fields from that daily integration are not used and instead the model atmosphere is initialised using ERA-Interim. The sea-ice initial conditions for the next

cycle are the unaltered output of the previous daily integration. Then the cycle starts again and the coupled model runs for another 24 hours. During this integration the sea-ice component is affected by the ocean innovations and the new atmosphere initial conditions via the coupler.

The ACCESS-S1 hindcast set is made up of nine members created by perturbing the atmospheric fields only with a random field perturbation (Hudson et al., 2017) and runs for 217 days for the period 1990–2012 initialised at the first

of every month. The ACCESS-S2 hindcast set used in this study runs for the period 1981–2018. Ensemble members are created in the same manner as ACCESS-S1 members, however, due to computing cost limitations, only three members per forecast initialisation date were run for 279 days. Bigger ensembles were generated by aggregating several three-member ensembles initialised on successive days (Wedd et al., 2022). Here, we build a nine-member time-lagged ensemble from three consecutive three-member forecasts initialised at the first of every month and the

two previous days and run for 279 days. We analyse the ensemble mean hindcasts unless otherwise specified.

Anomalies for each hindcast set are taken with respect to their own climatology specific to each initialisation date and forecast lead time, for the period 1990–2012. This serves as a first-order correction of model bias and drift. For monthly means, we define "0 lead time months" as the monthly mean forecast of the same month of initialisation.

Besides sea-ice concentration, we also analyse mean sea-ice thickness, which we compute as total sea-ice volume

divided by total sea-ice area.

## 2.2 Verification datasets

For verification we use satellite-derived sea-ice concentration, which estimates the proportion of each grid area that is covered with ice. Datasets derived using different algorithms and satellite platforms, each have their own biases and uncertainties. Estimates of inter-product uncertainty of sea-ice extent (SIE, defined here as the total region of the

Southern Ocean with at least 15% sea-ice cover) are of the order of 0.5 million $km^2$ (Meier and Stewart, 2019). As will be shown below, this spread is minimal compared with the typical errors in the ACCESS-S2 and ACCESS-S1 forecasts, so the overall conclusions of this study are independent of the verification dataset used.

We use NOAA/NSIDC's Climate Data Record V4 [CDR; Meier et al. (2014)] as the primary sea-ice verification dataset. It takes the maximum value of the NASA Team (Cavalieri et al., 1984) and NASA Bootstrap (Comiso,

2023) sea-ice concentration products to reduce their low concentration bias (Meier et al., 2014, 2021). Both source algorithms use data from the Scanning Multichannel Microwave Radiometer (SMMR) on the Nimbus-7 satellite



and from the Special Sensor Microwave/Imager (SSM/I) sensors on the Defense Meteorological Satellite Program's (DMSP) -F8, -F11, and -F13 satellites. The data have a spatial resolution of 25 by 25 km and daily from November 1978 onwards.

The European Organisation for the Exploitation of Meteorological Satellites (EUMETSAT) Ocean and Sea Ice Satellite Application Facility [OSI; EUMETSAT Ocean and Sea Ice Satellite Application Facility] based on the SSMIS sensor is another satellite-derived sea-ice concentration product. It is based on mostly the same sensors as the NOAA CDR but computed independently using different algorithms. Figures prepared with this dataset are provided in the appendix and do not differ significantly from the ones prepared using CDR.

## 2.3    Error measures

For evaluation purposes, we use a series of measures. Sea-ice extent is defined as the area of the ocean at least 15% covered by sea-ice. This threshold is motivated by the limitations in satellite retrieval, which is increasingly unreliable for lower sea-ice concentrations (Cavalieri et al., 1991).

Pan-Antarctic (net) sea-ice extent serves as a hemispheric measure of the amount of sea ice, but it does not take into 130 account the spatial distribution. A model could have a relatively accurate extent of the net ice but with different regional distributions. To account for location errors, we computed the Root Mean Squared Error (RMSE) of grid-point sea-ice concentration anomalies.

We compute RMSE as the square root of the area-averaged squared differences between grid-point forecasted and observed sea-ice concentration anomalies. We compute a pan-Antarctic RMSE by averaging over the whole 135 NOAA/NSIDC CDRV4 Southern Hemisphere domain, and also a zonally-varying RMSE computed over 15 longitude slices 24° wide around Antarctica.

All error measures were computed on the NOAA/NSIDC CDRV4 domain grid, to which model output was bilinearly interpolated. Note that the ACCESS CICE model grid has resolution between two and three times higher than NOAA/NSIDC CDRV4.

Forecast errors are also compared with hypothetical forecasts based on the persistence of anomalies and on climatology. The persistence forecast is generated by extending the observed sea-ice concentration anomalies the day of the forecast initialisation and comparing it with the actual anomalies observed. The climatological forecast error is computed as the standard deviation of daily anomalies.

As a measure of forecast improvement over the hypothetical forecast, we use the skill score (Murphy and Daan, 1985), 145 defined as

$$S = 1 - \frac{RMSE_f}{RMSE_r}$$





Where $RMSE_f$ is the RMSE of the forecast, $RMSE_r$ is the RMSE of the reference forecast. Negative skill score indicates that the forecast is worse than the reference forecast while positive values indicate an improvement. A perfect forecast would have zero RMSE and thus a skill score of 1.

## 2.4 Computational procedures

We performed all analyses in this paper using the R programming language (R Core Team, 2020), using data.table (Dowle and Srinivasan, 2020) and metR (Campitelli, 2020) packages. Significant processing was performed using the CDO command line operators (Schulzweida, 2023). All graphics are made using ggplot2 (Wickham, 2009). The paper was rendered using knitr and Quarto (Xie, 2015; Allaire et al., 2022).

# 3 Results and discussion

### 3.1 Bias

Figure 1 (Figure A1 for OSI) shows mean sea-ice extent of the ACCESS-S1 and ACCESS-S2 hindcasts (row a) and their differences from mean sea-ice extent of NOAA/NSIDC CDRV4 (row b). Mean extent at the first of every month is indicated with circles for the initial conditions and with triangles for the longest lead time possible for each
model (between 274 and 277 days for ACCESS-S2 and between 213 and 216 days for ACCESS-S1). At this long lead time, information of the initial conditions is essentially lost and the forecast reverts close to each model's preferred equilibrium state.

ACCESS-S2 initial conditions (circles in Fig. 1 column 2) show an overall negative bias, especially in the late summer-early autumn, while ACCESS-S1 initial conditions (circles in Fig. 1 column 1) are very close to observations,
as expected from the assimilation of sea-ice observations to produce the initial conditions of ACCESS-S1. Both systems' equilibrium states (triangles) show negative biases of sea-ice extent, particularly in the growth phase of late-autumn and winter months. This is due primarily to the melt season being longer than in observations and with faster melt between January and March and the growing seasons being shorter with slower growth during March and April. This is then followed by faster growth between May and July (Figure 2 and Figure A2). Many sea-ice models
exhibit this systematic underestimation during the sea-ice minimum and early freezing season (Massonnet et al., 2023), which could indicate problems in the representation of thermodynamics in the model (Zampieri et al., 2019). It is also not surprising that both forecasting systems converge to a similar equilibrium state because they share the same model formulation.

The difference between the initial conditions (circles) and the model equilibrium state (triangles) can be mostly
attributed to the effect of data assimilation, which in ACCESS-S2 is due solely to the coupling of sea-ice with





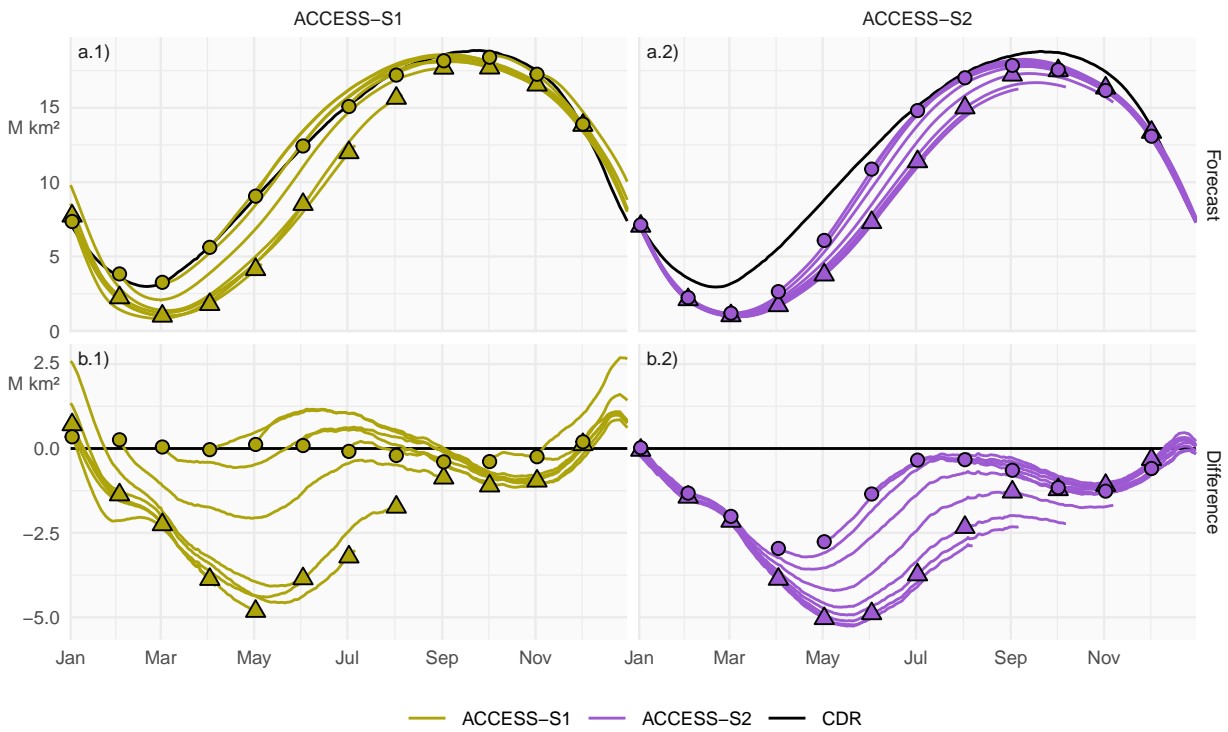

**Figure 1.** Row a: Pan-Antarctic daily mean sea-ice extent for all hindcasts initialised on the first of each calendar month for ACCESS-S1 (column 1; green) and ACCESS-S2 (column 2; purple). Observed mean sea-ice extent in each corresponding hindcast period is shown in black. Row b: Mean differences between the forecast and the observed values. Circles represent the initial conditions at the start of forecasts (i.e., the first of every month), and triangles represent the mean values at the lead time corresponding to the maximum lead time in S1 (between 213 and 216 days, depending on the month)

the atmosphere and the ocean. From May to October, in ACCESS-S2 circles are closer to observations than the triangles are, indicating that the information from the ocean and atmosphere data assimilation is affecting sea ice and improving the initial conditions. During these months, ACCESS-S1 can overestimate the sea-ice extent at short lead time. For the rest of the year circles are overlaid with triangles in ACCESS-S2, indicating that the ocean and 180 atmosphere data assimilation is not affecting sea ice and that this component of the model is virtually free-running.

To further understand the bias in ACCESS-S2, Figure 3 (Figure A3) shows spatial patterns of the differences of monthly mean sea-ice concentrations between NOAA/NSIDC CDRV4 and ACCESS-S2 hindcasts at the shortest monthly lead time. From October to May, the model underestimates sea-ice concentrations in most regions except for the inner Weddell Sea in April and May, where sea-ice concentrations saturate to 1 both in the observations and 185 forecasts. In winter, the differences are limited to a narrow band around the sea-ice edge with slight positive biases in





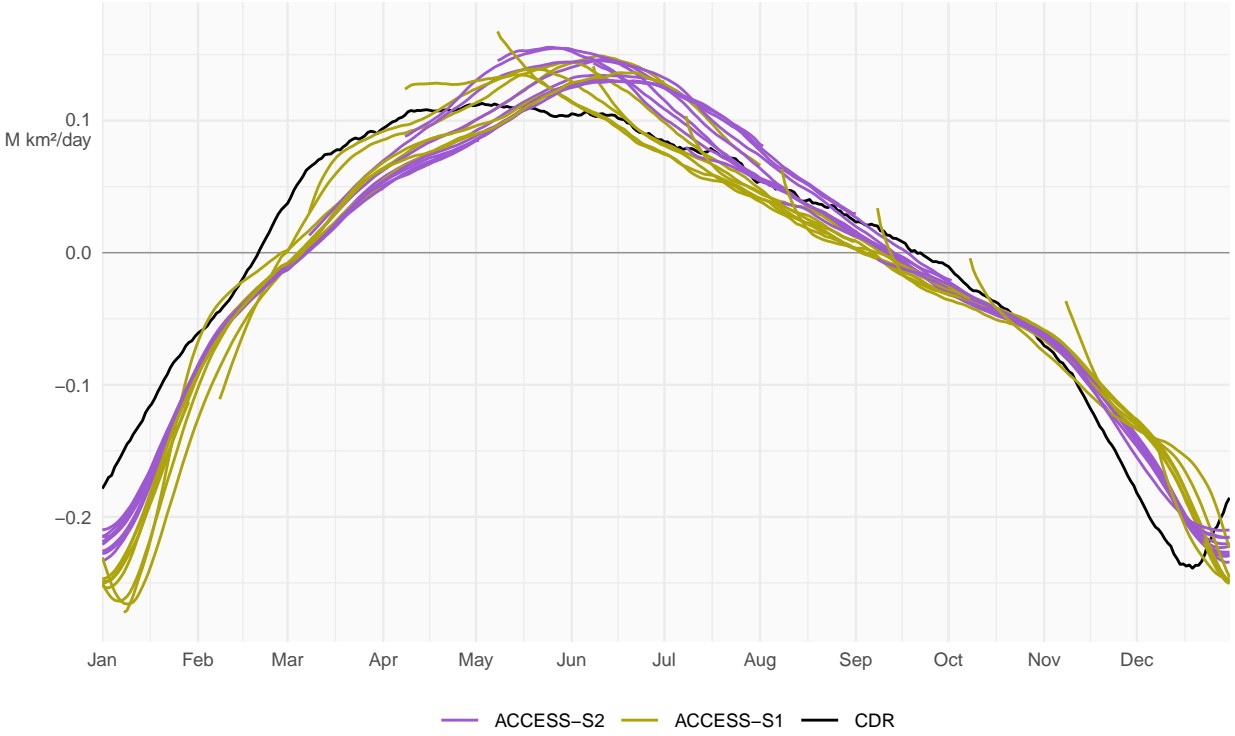

**Figure 2.** Mean daily sea-ice extent growth ($10^6 km^2/day$) in ACCESS-S1 (green) and ACCESS-S2 (purple) hindcasts and observations (black), computed as the mean daily differences in sea-ice extent between each date and the next for each forecast month. Values are smoothed with a 11-day running mean.

the African sector of East Antarctica and negative biases around the Indian Ocean sector which partially compensate, resulting in the near-zero extent bias seen in those months (Figure 1).

ACCESS-S1 has a comparatively smaller overall bias (Figure 4 and Figure A4). The largest values are found between April and June, when the faster growth results in large positive bias along the sea-ice edge, and in January, when the faster melt leads to large negative bias in the Weddell and Amundsen Seas.







**Figure 3.** Ensemble mean difference between monthly sea-ice concentration of ACCESS-S2 ensemble mean forecast at 0-month lead time (monthly mean values forecasted from the forecast initialised at the first of the month) and observations (CDR).





**Figure 4.** Same as Figure 3 but for ACCESS-S1.





## 3.2 RMSE



**Figure 5.** Monthly mean sea-ice extent anomalies of the observations (black) and forecasts from ACCESS-S1 (right column; purple) and ACCESS-S2 (left column; green) at lead times of 0, 2, 4, and 6 months. The RMSE and correlation during the overlapping period of ACCESS-S1 and ACCESS-S2 hindcasts (1990–2013) are shown on the top left and bottom left of each panel respectively.

Figure 5 (Figure A5) shows monthly sea-ice extent anomalies forecasted at selected lead times. Compared with ACCESS-S1, ACCESS-S2 anomaly forecasts are relatively poor (large RMSE) even for the first month (lead time 0),




whereas ACCESS-S1 forecasts stay relatively skilful even at a lead time of three months. ACCESS-S2 shows much
larger interannual variability than observations, with dramatic lows between 1995 and 2007, and highs between 2007
and 2015.

Unexpectedly, for ACCESS-S2, RMSE improves with lead time, even though the correlation degrades with lead time.
This is puzzling behaviour that goes contrary to what is usually seen in prediction models. The explanation seems to
be the mentioned increased interannual variability. Figure 6 (Figure A6) shows the interannual standard deviation of
monthly sea-ice extent of the forecasts as a function of lead time compared with observations. ACCESS-S1 standard
deviation lies within the observed standard deviation regardless of lead time, while ACCESS-S2 standard deviation is
more than twice that of observations at zero lead time and only approaches the observed value at nine month lead
time for most months.

ACCESS-S2 forecasts of sea-ice extent anomalies seem to align moderately well with observations (leading to
moderately high correlation) but their magnitude is overestimated (leading to large errors). This could be caused by
ACCESS-S2 sea ice being much more sensitive to atmospheric and oceanic forcing, perhaps due to lower thickness.

As an example, Figure 7 shows sea-ice concentration anomalies (top row) and sea-ice thickness and the difference
between the two models (bottom row) for 2 May 2008 initialised one day prior; being that close to initialisation
date, these are very approximately the initial conditions. ACCESS-S1 sea-ice concentrations anomalies are very close
to observations as expected from the system assimilating these data. ACCESS-S2 sea-ice concentration anomalies,
which are not assimilated, are not as close, but the large-scale pattern is aligned with observations. The system
simulates large positive anomalies in the Weddell and Ross Seas and slight negative anomalies in the Amundsen and
Bellingshausen Seas. The fact that ACCESS-S2 can simulate this pattern without assimilating sea-ice data suggests
that atmospheric and oceanic forcing were the dominant drivers. However, the magnitude of the sea-ice anomalies is
too big. It is plausible that this is due to the thinner ice simulated by ACCESS-S2 (bottom row).

Extending beyond the one case in Figure 7, Figure 8 shows monthly mean sea-ice thickness as a function of lead time
for ACCESS-S1 and ACCESS-S2. Supporting the idea that thinner ice is what causes the increased extent variability
in ACCESS-S2, this system simulates thinner sea-ice compared to ACCESS-S1 overall at almost all lead times and in
all months except for summer at short lead times (Dec-Jan, 0-1 months; Feb-Mar, 0-2 months). However, in both
systems, forecasted sea-ice is thicker at shorter lead times and then decreases, particularly in the summer months. If
thinner ice were a sufficient cause of increased variability, then we would expect variability to increase with lead time
in both forecasting systems.

The fact that ACCESS-S1 and ACCESS-S2 share the same model configuration and that the increased variability is
more extreme at short lead times (Fig. 6) suggests that the data assimilation procedure is partly responsible. It is
possible that sea-ice in the ACCESS-S2 system is left in an unbalanced state after assimilating atmospheric and





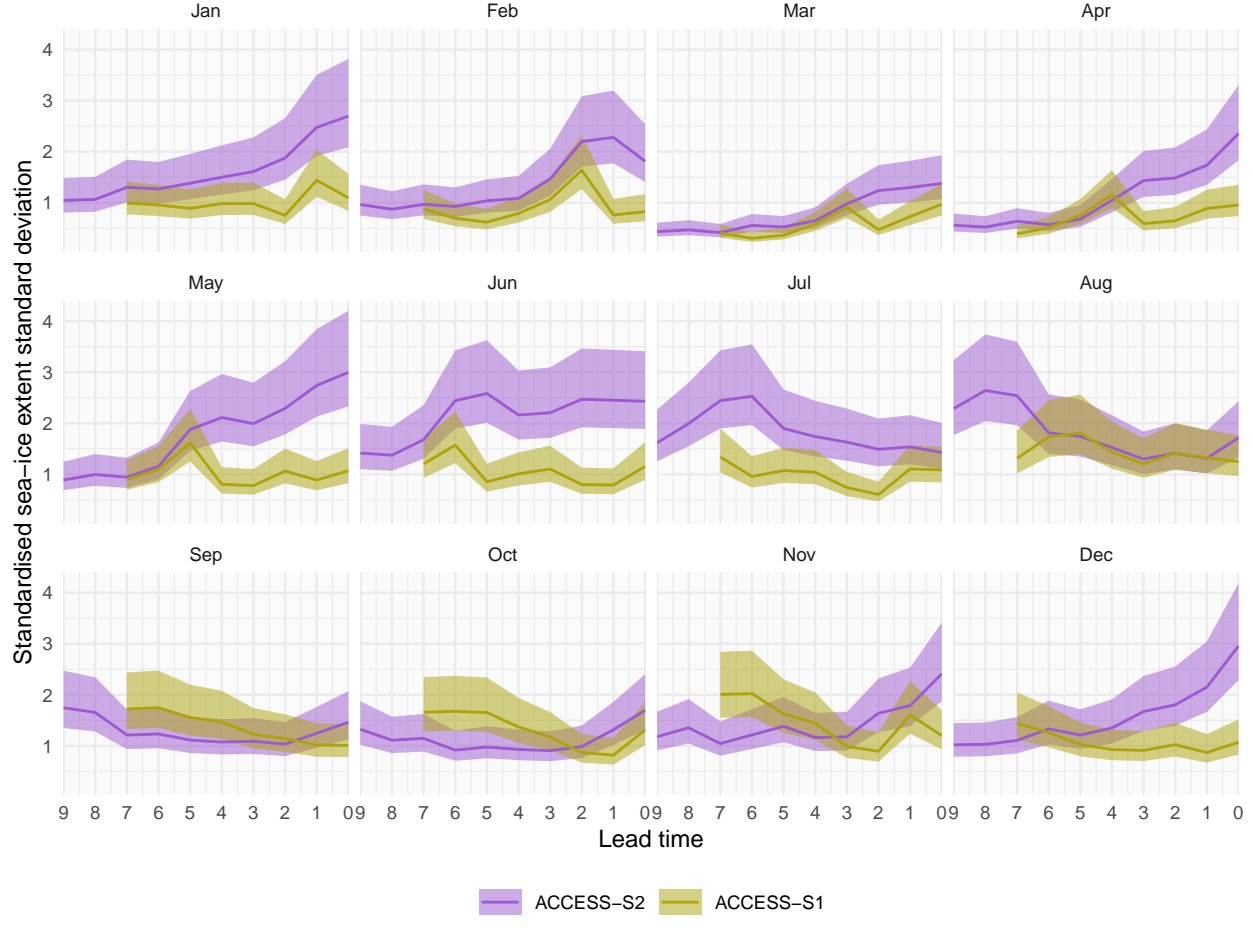

**Figure 6.** Interannual standard deviation with 95% confidence interval of monthly mean sea-ice extent forecasted for each month divided by that month's sea-ice extent observation standard deviation. ACCESS-S1 and ACCESS-S2 at different lead times. Each panel indicates the target month. Note the reverse horizontal axis.

oceanic data but not sea-ice data, leading to large responses that are amplified by the thin ice in the initial states which then subside at longer lead times when the model is balanced.

To assess ACCESS-S2 forecasts in more detail, we compute error measures for all hindcasts started on the 1st of every month. Figure 9 (Figure A7) shows the mean RMSE of sea-ice concentration anomalies for ACCESS-S1 and ACCESS-S2 hindcasts compared against persistence and climatological forecasts used as a benchmark. Due to errors in the initial conditions, it is expected that persistence forecasts would be better than the model forecasts at very short lead times, but that the persistence forecast errors would grow faster and may eventually surpass the model forecast errors. The black line shows that the persistence forecast error indeed grows rapidly and reaches its maximum







**Figure 7.** ACCESS-S1 and ACCESS-S2 hindcasts for 2 May 2008 at one day lead time. Top row shows sea-ice concentration anomalies forecasted by each system and the observations. Bottom row shows forecasted sea-ice thickness and the difference between ACCESS-S1 and ACCESS-S2.

in about 30 days for most months except for February, when it grows much slower. The ACCESS-S1 forecast errors grow slower than persistence forecast errors and remain lower after less than 10 days on average. The ACCESS-S2 forecast error starts high in all months and is lower than the persistence forecast error after more than 15 days in most months except for forecast initialised in February, when it takes 80 days.




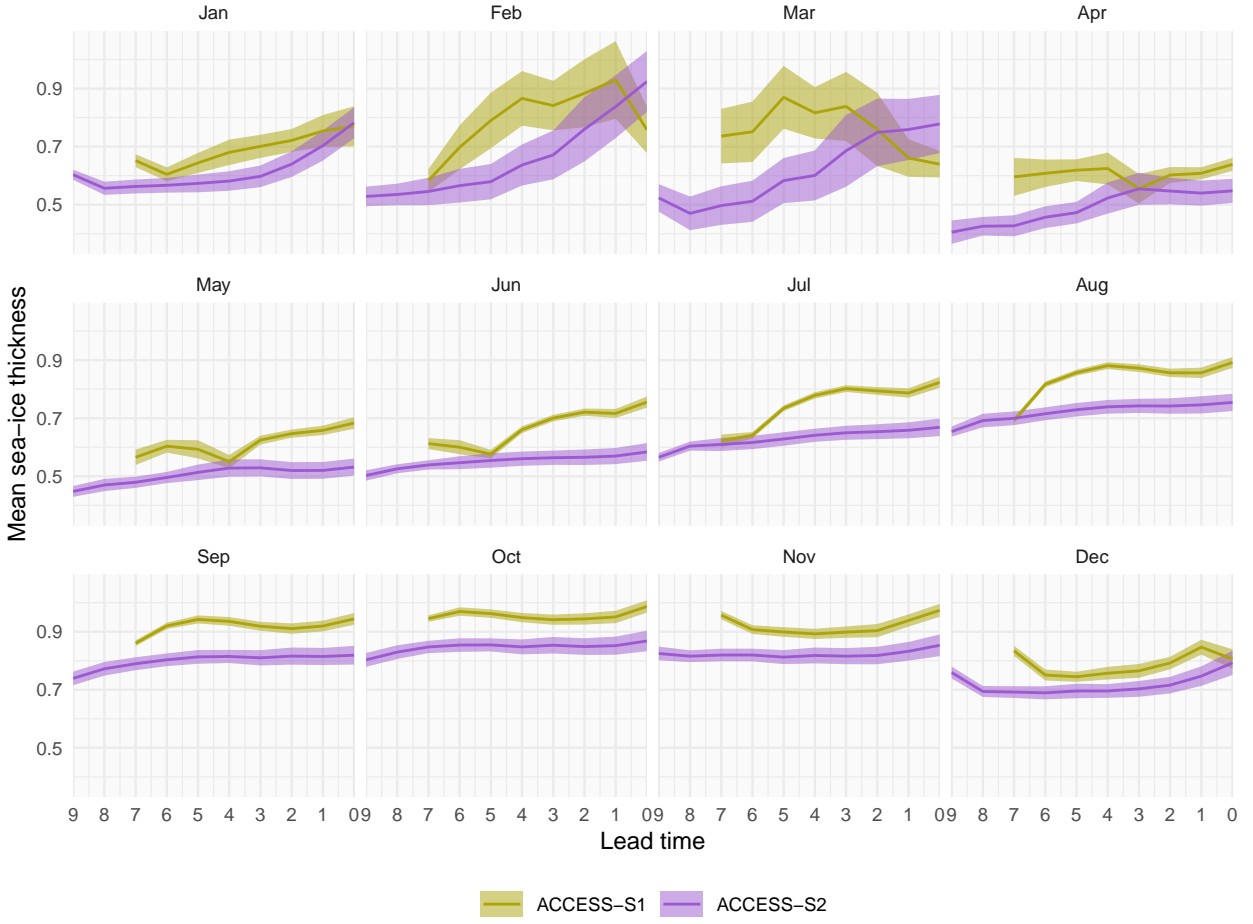

**Figure 8.** Mean and 95% interval of monthly mean sea-ice thickness for ACCESS-S1 and ACCESS-S2 at different lead times. Each panel indicates the target month. Note the reverse horizontal axis.

At longer lead times, it is more appropriate to compare errors with the climatological forecast error. The lead time at which ACCESS-S1 forecast error is higher than the climatological forecast error varies between more than 60 and less than 20 days depending on forecast initialisation month with the minimum in June. ACCESS-S2 forecasts never have lower error than climatology, on the other hand, except marginally in October forecasts.

Figure 10 (Figure A8) summarises the lead time window in which each hindcast is better than both the persistence forecast and the climatological forecast as a function of forecast month. ACCESS-S1 forecasts have a wider lead time window in the summer than the other seasons and is not better than both benchmarks at forecasting June sea-ice concentration anomalies. Forecasts initialised in May and June are particularly poor, and July cannot be forecasted



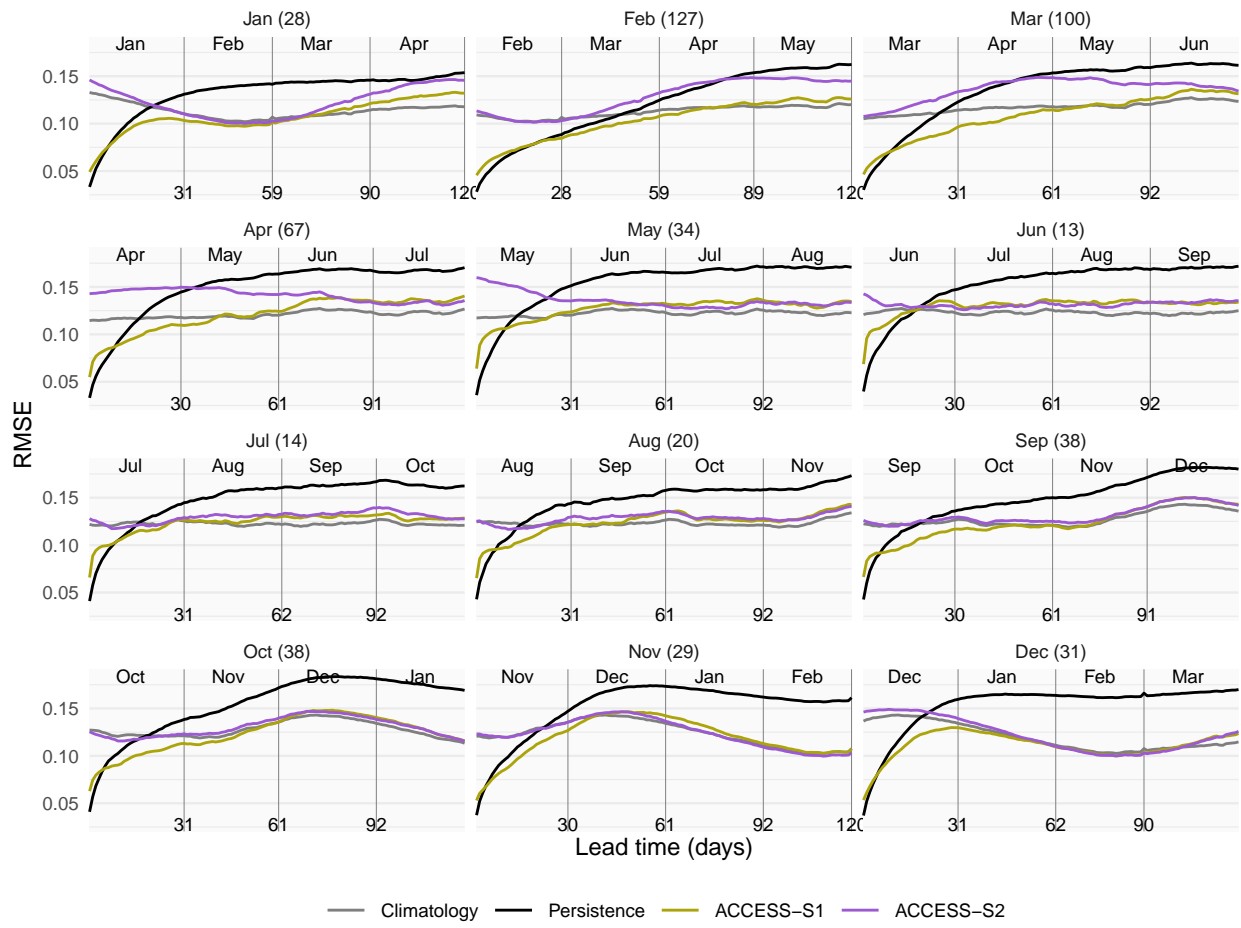

**Figure 9.** Mean RMSE of sea-ice concentration anomalies as a function of forecast lead time for all forecasts initialised on the first of each month compared with a reference forecast of persistence of anomalies (black) and climatology (gray). Only the first 120 days are shown. In parentheses, the shortest time at which ACCESS-S1 and ACCESS-S2 mean RMSE is not statistically different at the 99% confidence level.

better than the benchmarks. This is consistent with the mid-winter loss of predictability observed by Libera et al. (2022), who attributed it to deep warm water entraining into the mixed layer.

To analyse the spatial distribution of the model error, we computed the RMSE of zonal mean sea-ice concentration anomalies on 15 slices of 24° longitude span for each forecasting system. We control for some areas being naturally easier to forecast than others by computing the RMSE skill score with the climatological forecast RMSE as reference.

For ACCESS-S1 forecasts (Figure 11 and Figure A9), skill tends to be lower off the coast of Eastern Antarctica even at short lead times; for instance, the skill score for forecasts initialised in May and June are negative between 0° and




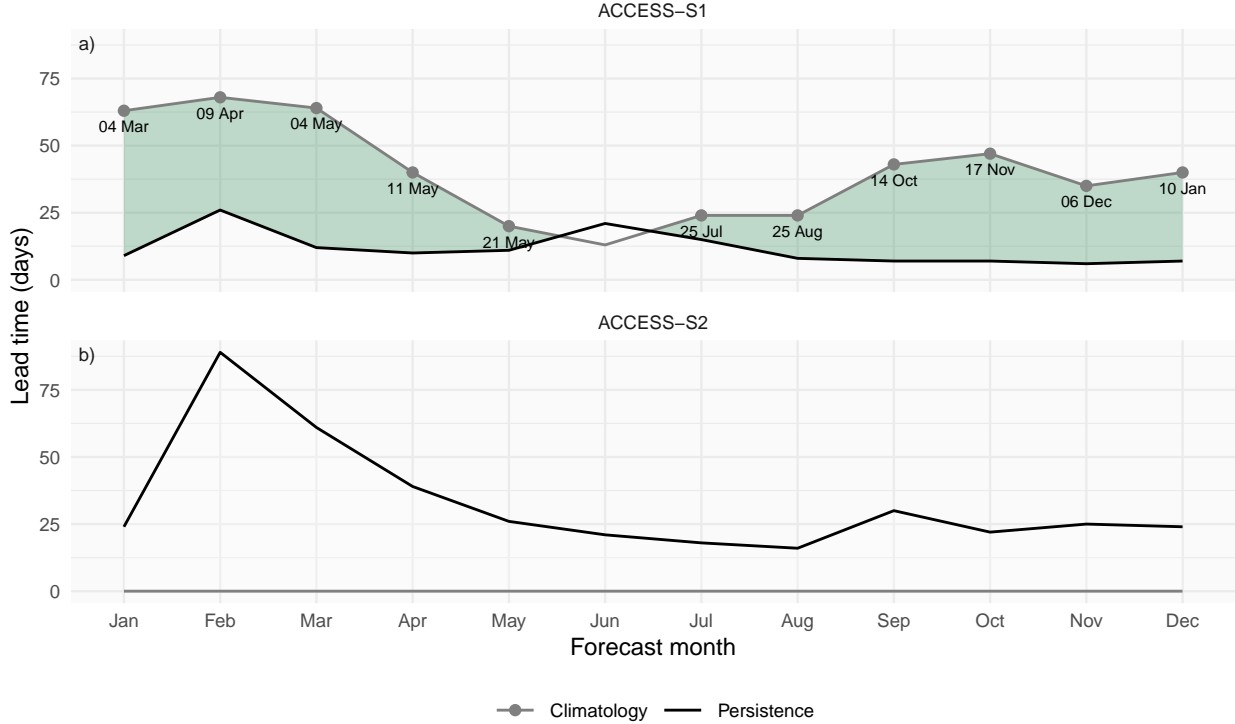

**Figure 10.** Minimum lead time at which each forecast's mean RMSE becomes larger than the lower bound of the 95% confidence interval of persistence forecast RMSE (black lines) and maximum lead time at which each forecast's mean RMSE remains lower than the lower bound of the 95% confidence interval of climatological forecast RMSE (gray lines). Green shading indicates the window where forecasts outperform both persistence (lead times longer than black line) and climatology (lead times shorter than gray line). Text labels show the date corresponding to the maximum lead time at which each forecast outperforms climatology.

120°E even at almost zero lead time. This mirrors Libera et al. (2022) findings of a "winter predictability barrier", although they focus on the Weddell Sea and here we show that the effect seems to be stronger more to the east. In
255 West Antarctica there is a hint of easterly-propagating skill in forecasts initialised in February and March. This is consistent with Holland et al. (2013) findings that memory of sea-ice anomalies are stored in ocean heat content anomalies that are transported east by the Antarctic Circumpolar Current.

ACCESS-S2 forecasts (Figure 12 and Figure A10) also have lower skill over East Antarctica. From July to December even though the pan-Antarctic average skill is negative at all lead times (Fig. 10), it is positive for up to a month in
260 West Antarctica. Since oceanic and atmospheric forcing is the only source of information, this suggests that sea-ice in this region is particularly sensitive to oceanic and atmospheric forcing and suggests a role of the Pacific-South American mode and the Amundsen Sea Low to shape sea-ice concentration anomalies. The fact that this is evident





**Figure 11.** RMSE skill score of ACCESS-S1 forecasts with climatological forecast as reference computed on 15 meridional slices 24° wide as a function of lead time and longitude. Antarctica's coastline is shown at the bottom of each panel for reference.



**Figure 12.** Same as Figure 11 but for ACCESS-S2.

in the months in which El Niño–Southern Oscillation teleconnections are more important for atmospheric circulation





also suggests the influence of tropical Pacific variability. February and March are the only two months that can be forecasted with marginally positive skill in large regions.

**Figure 13.** Same as Figure 11 but for the difference between ACCESS-S1 and ACCESS-S2.



Finally, Figure 13 (Figure A11) shows the difference in skill between ACCESS-S1 and ACCESS-S2. Large differences in skill indicate areas and months that are most affected by the data assimilation present in ACCESS-S1. Between January and March, which are the months in which ACCESS-S1 is the most skilful (Figure 10), most of the improvement compared with ACCESS-S2 is present in the Ross and Weddell Sea. In April and May, the improvement seems more homogeneous.

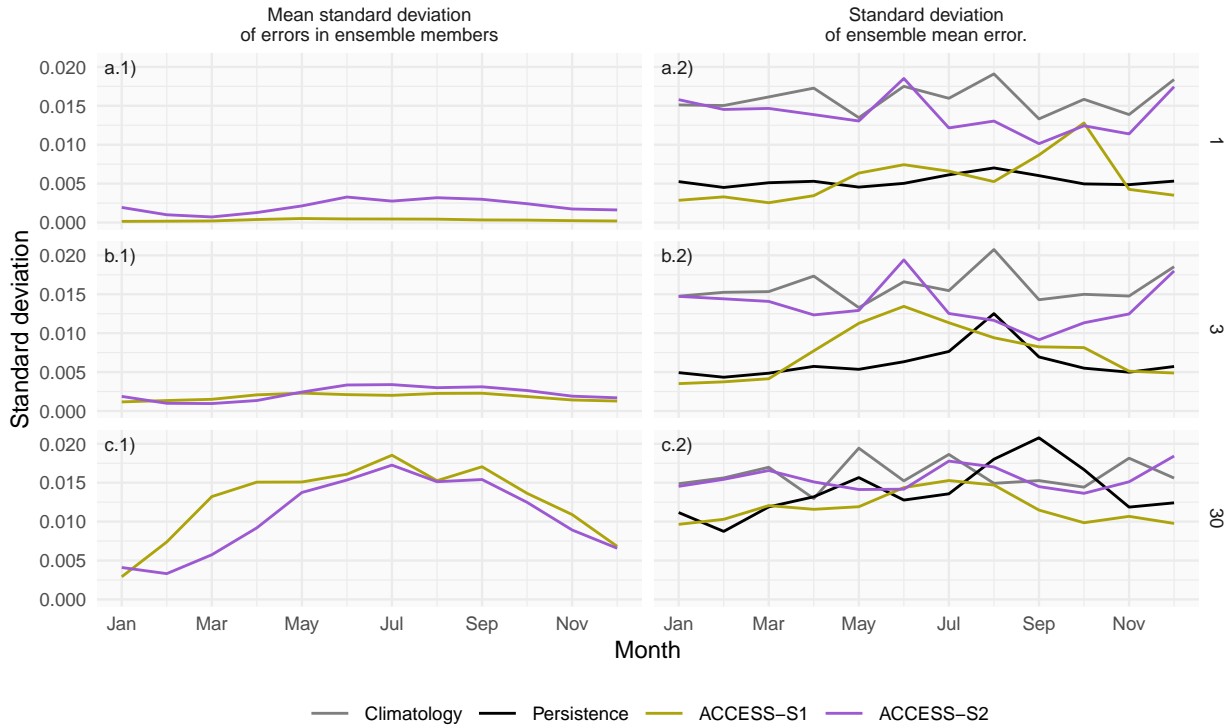

**Figure 14.** Decomposition of forecast error spread at 1, 5 and 30 days lead time for ACCESS-S1 and ACCESS-S2 hindcasts across initialization months. The left column shows the mean standard deviation of RMSE errors across ensemble members, while the right column shows the standard deviation of the ensemble mean RMSE error and the spread of the persistence and climatology forecasts errors.

In Figure 9 the mean error was shown. Figure 14 (Figure A12) column 1 shows the mean standard deviation of errors among ensemble members at various lead times. At one day lead time (Fig. 14 a.1) ACCESS-S2 has a slightly larger spread than ACCESS-S1 due to the way that ensemble members are generated. ACCESS-S1 ensemble members are generated by adding random field perturbations to the atmosphere only, which then are transferred to the other components via the coupled simulation (Hudson et al., 2017). With this scheme, ensemble members are all but guaranteed to be underdispersed in the ocean and sea-ice components. The time-lag ensemble used for ACCESS-S2





ensures greater spread. This difference is gone after about just two days, and both systems have a comparable spread in ensemble member error afterwards (Fig. 14 b1 and c1).

Figure 14 column 2, on the other hand, shows the standard deviation of ensemble mean error of each hindcast and the persistence forecast. At one day lead time, ACCESS-S2 ensemble mean error standard deviation is much larger than ACCESS-S1's, which in turn is comparable to the persistence forecast error standard deviation. At longer lead times, the spread of ACCESS-S1 and persistence forecast standard deviation increases to eventually be comparable to ACCESS-S2 and the standard deviation in climatological forecast errors. ACCESS-S2 error standard deviation is fairly independent of lead time and similar to the climatological forecast error standard deviation at all lead times.

## 3.3 Conclusions

Sea-ice forecasts from the ACCESS-S2 system show a significant low extent bias, particularly during late summer and early autumn. This bias is attributed to a faster and longer melt season between January and March, and slower growth between March and April. This underestimation during the minimum and early freezing season is a common issue in many seasonal-to-subseasonal (S2S) systems, suggesting potential problems either with the model's thermodynamic representation or with short wave radiation forcing, as shown in other climate models (Zampieri et al., 2019; Roach et al., 2020). Even though ACCESS-S2 shares the same model components as ACCESS-S1, the latter does not suffer from this bias, indicating that assimilating sea-ice concentrations successfully corrects for the negative bias that exists in the free-running model.

Ensemble spread grows quickly even when perturbations are only implemented in the atmosphere component (in ACCESS-S1), indicating that sea ice is indeed responding quickly to atmospheric perturbations. However, our analysis suggests that the atmosphere and ocean data assimilation implemented in ACCESS-S2 is only effectively influencing sea-ice initial conditions from June to October, while the rest of the year, the sea-ice component runs virtually free, reverting to its biased equilibrium state. Zhou and Alves (2022) had previously evaluated sea-ice forecasts in ACCESS-S2 and also highlighted the poor performance of this forecasting system attributed to the lack of good initial conditions.

Analysis of the error spread shows that ACCESS-S2 initial conditions from December to May not only have large errors, but that the initial error spread is very large compared with ACCESS-S1. This spread is not due to the perturbation scheme, since the mean error variance for individual forecasts is low and comparable with ACCESS-S1. Instead, it is due to large variance of the mean error of individual forecasts, which is comparable to the climatology spread. This is further evidence that individual initial conditions are not being affected by the data assimilation scheme.

Although ACCESS-S1 only assimilates sea-ice concentration, it is clear that sea-ice thickness is also affected through the assimilation process. ACCESS-S1 simulates significantly thicker ice than ACCESS-S2 and in both systems sea-ice



is thicker at shorter lead times than at longer lead times. Both the explicit data assimilation in ACCESS-S2 and the effects of atmospheric and oceanic data assimilation in ACCESS-S1 might be nudging simulated sea ice to be thicker than the model equilibrium state. We suggest that the thinner sea ice in ACCESS-S2 contributes to the large sea-ice extent variance, but other mechanisms, such as unbalanced initial conditions might also be important.

Given that ACCESS-S2 sea-ice extent is not directly initialised by sea-ice observations, comparing its forecasts with those of ACCESS-S1 allows us to estimate the time-scale over which initial conditions are important. We find that initial conditions affect Antarctic sea-ice forecasts in the order of a few months, but that effect is seasonally dependent. January to April initial conditions improve forecasts for up to three months. February initial conditions in particular are shown to be crucial for determining sea-ice evolution at least up to May. Arctic sea-ice forecasts also show greater sensitivity to initial conditions in boreal summer, compared with boreal winter (Day et al., 2014; Bunzel et al., 2016), suggesting a similar mechanism might be playing a role.

Forecasts initialised in winter have very little skill and ACCESS-S1 and ACCESS-S2 forecast errors are statistically indistinguishable after just two weeks. This is consistent with Libera et al. (2022)'s finding of a "winter predictability barrier" in the Weddell Sea, although they describe the barrier as a sharp loss of predictability in July, and here we find a gradual reduction in skill compared with climatology around June. This difference might be due to our use of pan-Antarctic RMSE, since our regional analysis indicates that the degraded skill is most dramatic in the King Haakon Sea.

These findings have important implications for both operational forecasting, model development and predictability studies. For operational centers, our results suggest that efforts to improve sea-ice data assimilation should prioritize the summer and autumn months when initial conditions have the greatest impact on forecast skill. Additionally, the substantial bias in ACCESS-S2 highlights the need for improved model physics, particularly in the representation of sea-ice thermodynamics and radiation processes. Crucially, our results suggest dramatic seasonal variations in sea-ice predictability. Future studies should therefore use initial conditions through the whole year rather than focusing on a limited number of initialisation dates.

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



# 5 Appendix

The following are the same figures from the main paper but using the OSI dataset instead of CDR.

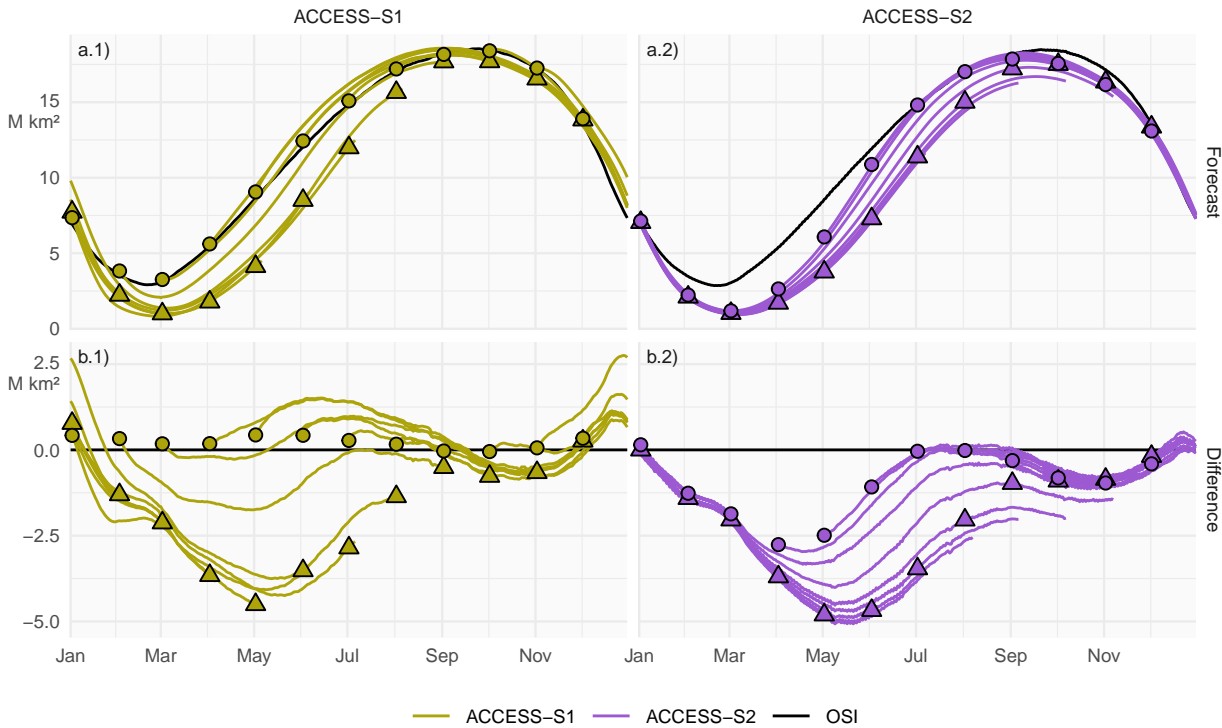

**Figure A1.** Row a: Pan-Antarctic daily mean sea-ice extent for all hindcasts initialised on the first of each calendar month for ACCESS-S1 (column 1; green) and ACCESS-S2 (column 2; purple). Observed mean sea-ice extent in each corresponding hindcast period is shown in black. Row b: Mean differences between the forecast and the observed values. Circles represent the initial conditions at the start of forecasts (i.e., the first of every month), and triangles represent the mean values at the lead time corresponding to the maximum lead time in S1 (between 213 and 216 days, depending on the month)



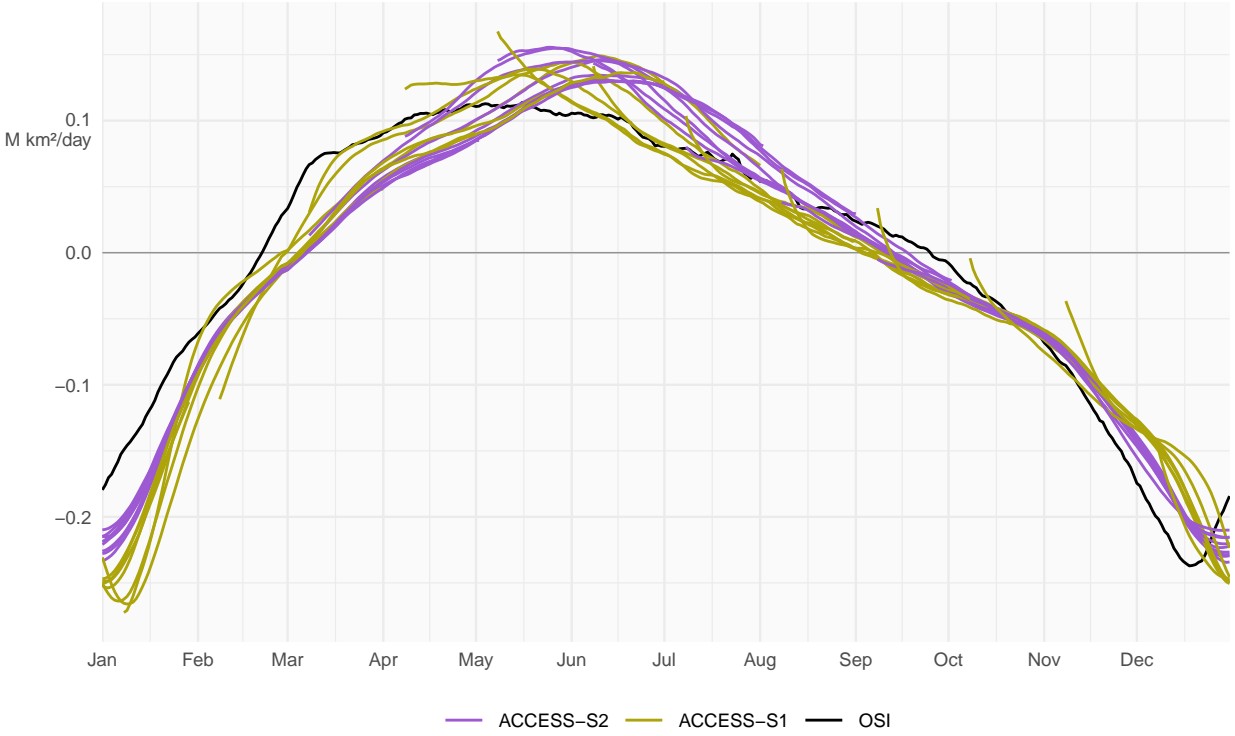

**Figure A2.** Mean daily sea-ice extent growth ($10^6 km^2/day$) in ACCESS-S1 (green) and ACCESS-S2 (purple) hindcasts and observations (black), computed as the mean daily differences in sea-ice extent between each date and the next for each forecast month. Values are smoothed with a 11-day running mean.



**Figure A3.** Ensemble mean difference between monthly sea-ice concentration of ACCESS-S2 ensemble mean forecast at 0-month lead time (monthly mean values forecasted from the forecast initialised at the first of the month) and observations (OSI).





**Figure A4.** Same as Figure 3 but for ACCESS-S1.



**Figure A5.** Monthly mean sea-ice extent anomalies of the observations (black) and forecasts from ACCESS-S1 (right column; purple) and ACCESS-S2 (left column; green) at lead times of 0, 2, 4, and 6 months. The RMSE and correlation during the overlapping period of ACCESS-S1 and ACCESS-S2 hindcasts (1990–2013) are shown on the top left and bottom left of each panel respectively.





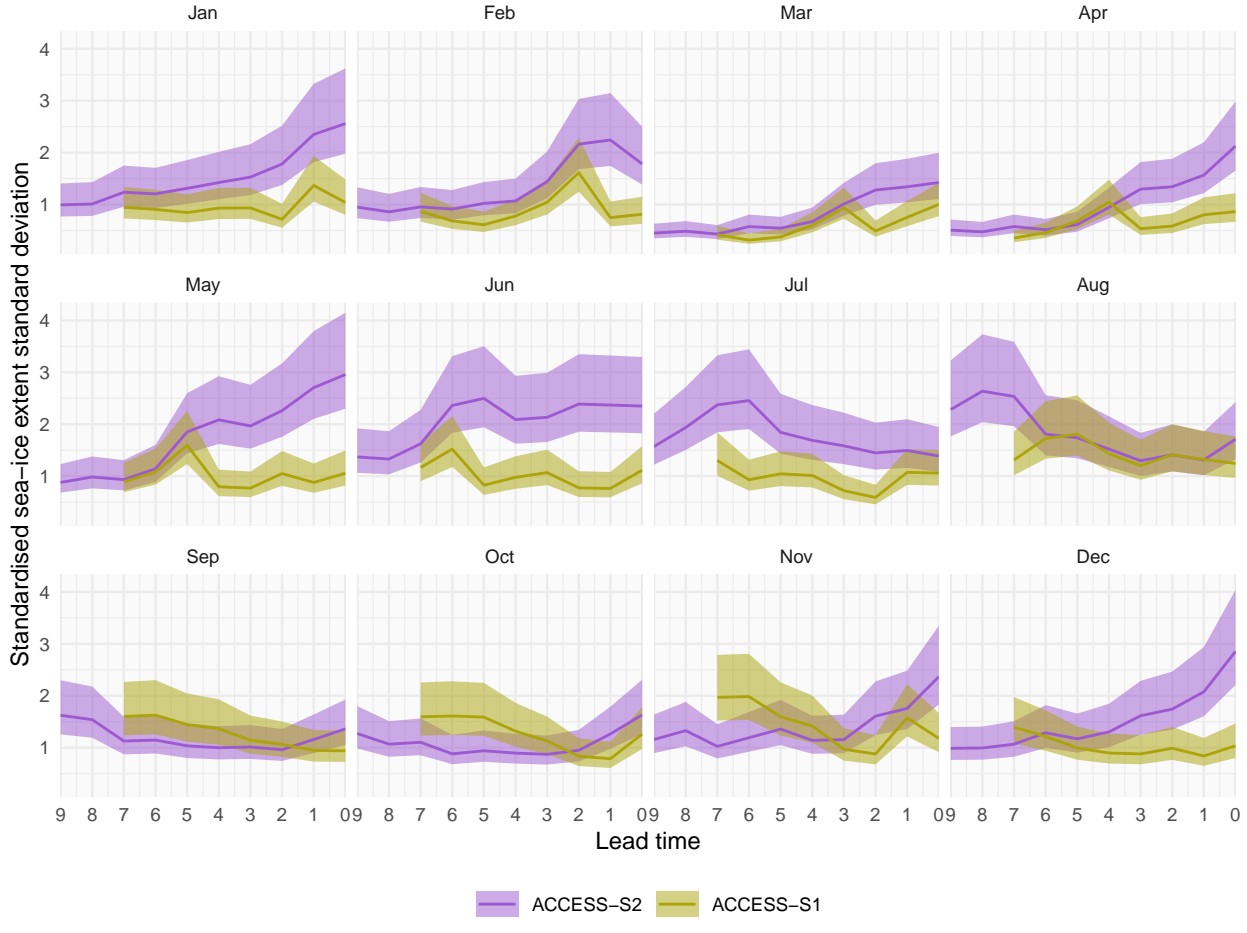

**Figure A6.** Interannual standard deviation with 95% confidence interval of monthly mean sea-ice extent forecasted for each month divided by that month's sea-ice extent observation standard deviation. ACCESS-S1 and ACCESS-S2 at different lead times. Each panel indicates the target month. Note the reverse horizontal axis.





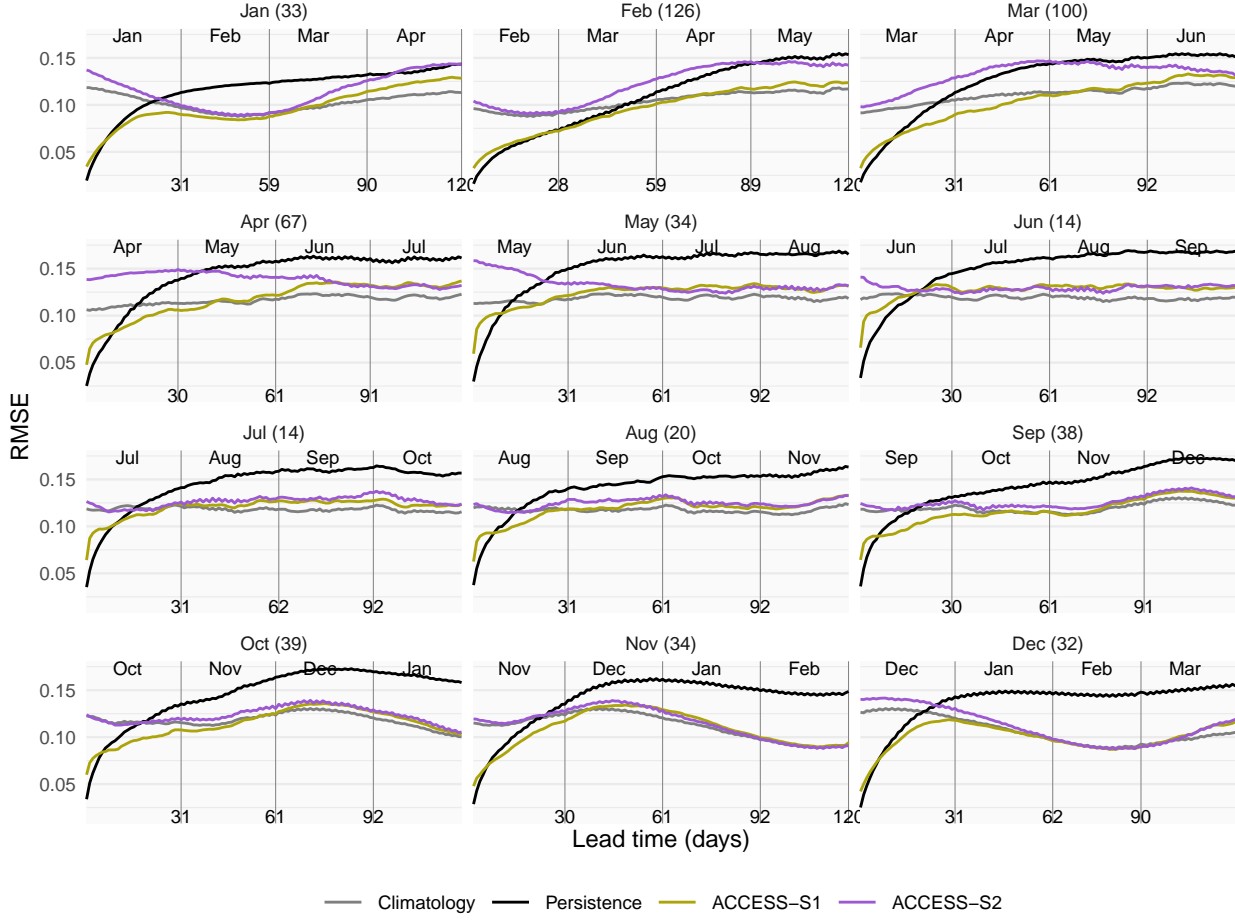

**Figure A7.** Mean RMSE of sea-ice concentration anomalies as a function of forecast lead time for all forecasts initialised on the first of each month compared with a reference forecast of persistence of anomalies (black) and climatology (gray). Only the first 120 days are shown. In parentheses, the shortest time at which ACCESS-S1 and ACCESS-S2 mean RMSE is not statistically different at the 99% confidence level.





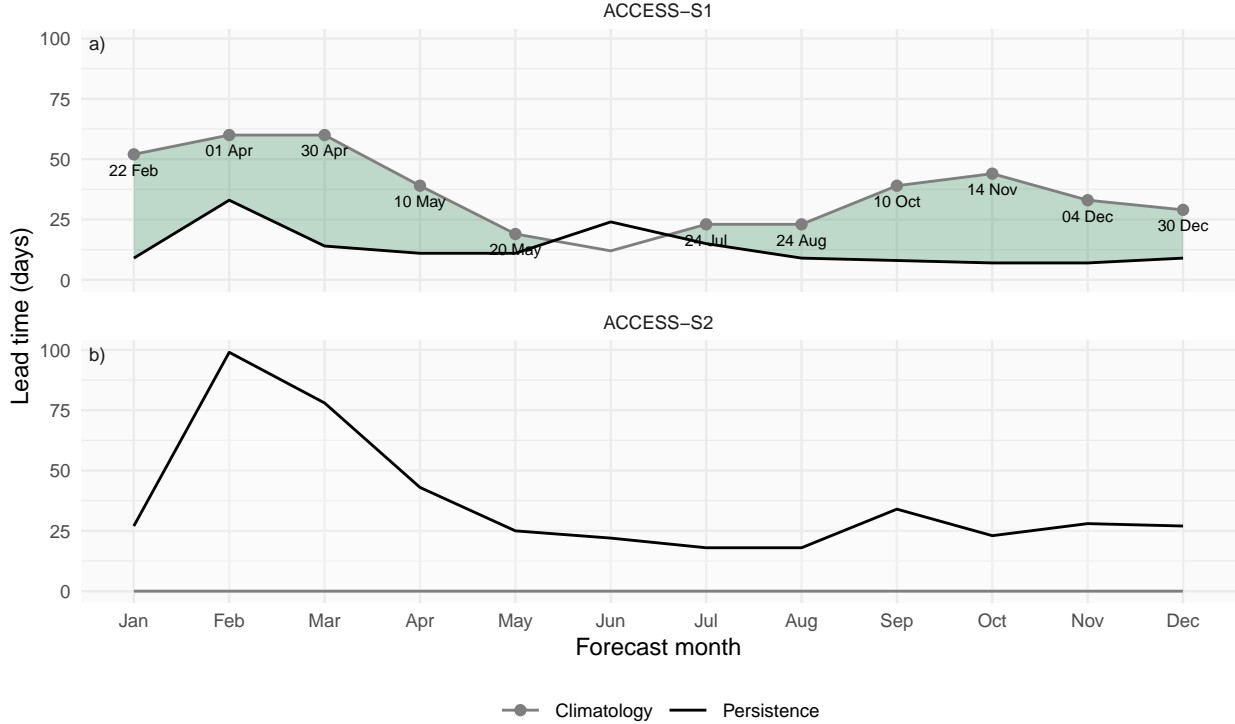

**Figure A8.** Minimum lead time at which each forecast's mean RMSE becomes larger than the lower bound of the 95% confidence interval of persistence forecast RMSE (black lines) and maximum lead time at which each forecast's mean RMSE remains lower than the lower bound of the 95% confidence interval of climatological forecast RMSE (gray lines). Green shading indicates the window where forecasts outperform both persistence (lead times longer than black line) and climatology (lead times shorter than gray line). Text labels show the date corresponding to the maximum lead time at which each forecast outperforms climatology.

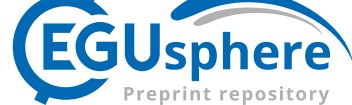



**Figure A9.** RMSE skill score of ACCESS-S1 forecasts with climatological forecast as reference computed on 15 meridional slices 24° wide as a function of lead time and longitude. Antarctica's coastline is shown at the bottom of each panel for reference.




**Figure A10.** Same as Figure 11 but for ACCESS-S2.





**Figure A11.** Same as Figure 11 but for the difference between ACCESS-S1 and ACCESS-S2.




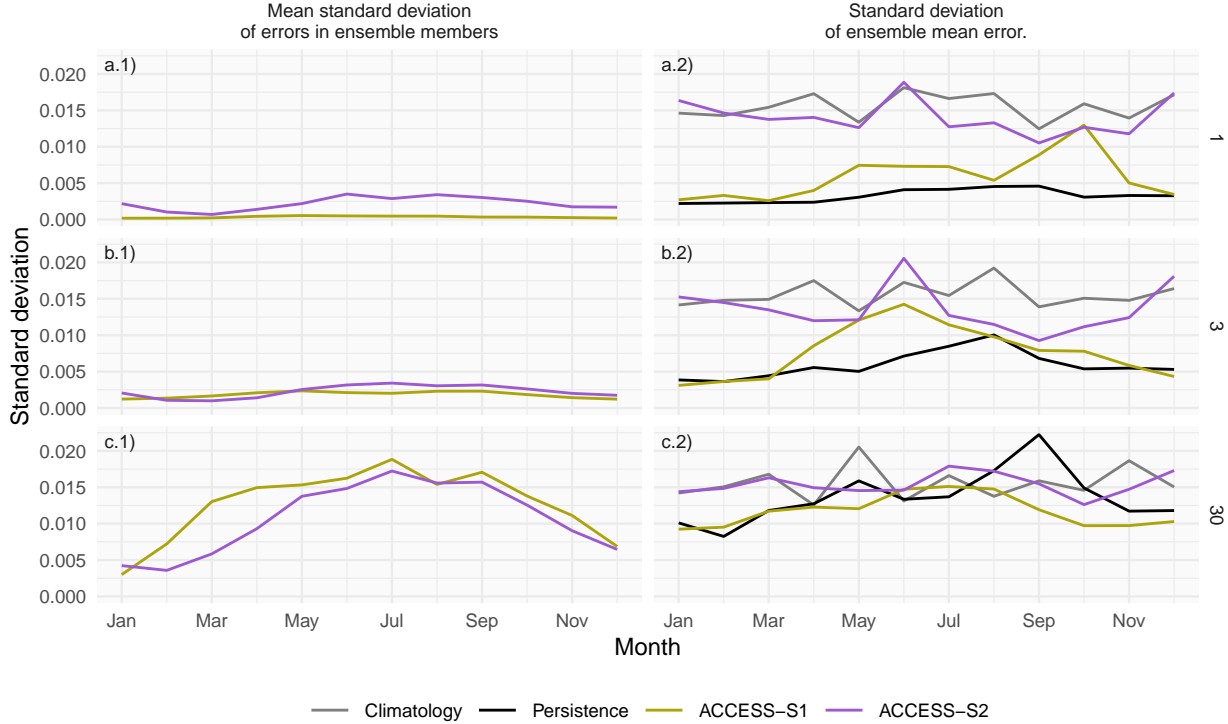

**Figure A12.** Decomposition of forecast error spread at 1, 5 and 30 days lead time for ACCESS-S1 and ACCESS-S2 hindcasts across initialization months. The left column shows the mean standard deviation of RMSE errors across ensemble members, while the right column shows the standard deviation of the ensemble mean RMSE error and the spread of the persistence and climatology forecasts errors.

### Code/Data availability

The underlying code for this study is available on GitHub: https://github.com/eliocamp/access-s2_ice-eval. Raw data of +S1 and +S2 hindcast are not available due to size. Derived datasets required to reproduce the results, including extent timeseries and error measures, are available in this Zenodo repository: https://zenodo.org/records/17479538 (Campitelli, 2025)

### Author contributions

EC performed the data analysis and wrote the manuscript draft. AP, JA, EL, MW and PR, performed interpretation of the 470    results, and reviewed and edited the draft. All authors read and approved the final manuscript.



## Competing interests

The authors declare no competing interests.

## Acknowledgements

We thank the internal reviewers Bethan White and Xiaobing Zhou for their comments and feedback. This work benefited from
earlier unpublished work by Laura Davies, Phil Reid, Andrew G. Marshall. This research was undertaken with the assistance
of resources from the National Computational Infrastructure (NCI Australia), an NCRIS enabled capability supported by the
Australian Government. This work was supported by ARC SRIEAS Grant SR200100005 Securing Antarctica's Environmental
Future.