# Peer review of "The Importance of Initial Conditions in Seasonal Predictions of Antarctic Sea Ice"

_EGUsphere, 2025_

## Referee Comment (RC1)

Review comment for 'The Importance of Initial Conditions in Seasonal Predictions of Antarctic Sea Ice' submitted to *The Cryosphere* (Manuscript ID: *EGUSPHERE-2025-6049*)

In this study, Campitelli et al. investigated the importance of initial condition on seasonal prediction of Antarctic sea ice. Two versions of Australian Bureau of Meteorology's ACCESS seasonal forecast system are employed, with one assimilating sea ice concentration (SIC) data (ACCESS-S1) while another not (ACCESS-S2). Based on more than 20 years hindcasts, the authors performed a detailed analysis of model climatology, interannual variability, and the prediction skill across different months and regions.

Overall, the author found that ACCESS-S1 exhibits better prediction skill than the ACCESS-S2, which is attributed to the SIC data assimilation, particular for summer- and autumn-initialized predictions. In contrast, the benefit of SIC assimilation is limited for winter-initialized predictions. The author also found that the ACCESS-S2 shows larger interannual variability of sea ice extent (SIE) than the ACCESS-S1 possibly due to its thinner ice, which is more sensitive to the atmospheric and oceanic forcing.

I find this manuscript to be generally well written, and the figures are clearly and carefully presented. However, several aspects requires further clarification and investigation, and the scientific novelty could be strengthened. I therefore recommend major revision to this manuscript. My detailed comments are listed as below.

General comments:

1) the authors quantify the importance of initial conditions on seasonal prediction by comparing two versions of ACCESS model. However, the data assimilation scheme of these two versions is different in multiple aspects. This raises the question of which components of the initial conditions are truly responsible for the skill differences. In this manuscript, the authors mostly examine the role of sea ice initial condition but the potential influence of other variables appears to be underexplored and should be discussed more explicitly.

2) It remains unclear whether the interannual variability of SIE increases with lead time in ACCESS-S2, while this behavior is not evident in ACCESS-S1. The authors suggest that the thinner ice in ACCESS-S2 is responsible for this difference; however, sea ice becomes thinner with lead time in both systems. The authors then propose that data assimilation may lead to an imbalanced state. Could the authors provide a clearer conclusion or more direct evidence to explain the contrasting behavior between ACCESS-S1 and ACCESS-S2?

3) Many climate models suffer from systemical bias, and bias correction is often applied prior to forecast evaluation. Have the authors examined whether bias correction would lead to substantial differences in the reported results? Clarifying the role of model bias in the skill assessment would strengthen the robustness of the conclusions.

Specific comment:

P5 Line146, I have a careful reading on the textbook by Murphy and Daan (1985) and found that the skill score **S** is defined based on the MSE rather than the RMSE (see equation 23 in the textbook), despite I believe these two definitions won't lead to dramatic difference in conclusion.

P7 Line 175, according to my observe, the period should be from April to September?

P12 Line 200, here the authors suggest that the decrease in interannual variability with increasing lead month in ACCESS-S2 is responsible for the improvement in RMSE. However, I found that for some months (e.g., July, August, September) the interannual variability increase with the lead time. Could the authors clarify how this explanation applies to these months?

P12 Line 206, how does the sea ice thickness adjust when assimilating the sea ice concentration?

P12 Line 210, please consider examining the role of the ocean in the bias of sea ice concentration magnitude. I understand that the ocean conditions are nudged toward a referrence dataset only when SST excceds 0 degree. It is therefore possible that, in the regions of interest, SST remains below this threshold, in which case the nudging wouldn't effectively remove ocean-related model biases.

P12 Line 211, which doesn't assimilate

P12 Line 214, sea ice concentration anomaly

P15 Line 245, should it be the June that cannot be forecasted better than the benchmarks? In addition, the metric employed in Libera et al (2022) was ACC, which is slightly different from the RMSE used in this study.

P17 Line 255, Is the eastward propagation feature also evident in December? Are there any criteria used to determine whether eastward propagation is present?

P22 Line 289, subseasonal to seasonal.

Figure

Figure 1: Could the author clarify why the maximum lead time in ACCESS-S1 varies between 213 and 216 days. Was this choice intended to align with the ACCESS-S2?

Figure 5: what does the word in the bracket represent, e.g., (CI: 0.04…0.27). The confidence interval? Then which confidence level? I recommend adding the asterisk to show the result is significant or not.

Figure 6, how do you compute the 95% confidence level? Is it based on an assumption of normal distribution for the ratio of SIE between the prediction and the reference?

Figure 9, I found that the RMSE of ACCESS-S2 mostly overlay with that of climatology. Does this imply that the oceanic and atmosphere data assimilation provide limited benefit to the prediction? What role does the model bias play in this prediction error?

Figure 10-11, please consider clarifying whether this skill score is significant or not

---

## Referee Comment (RC2)

**Review to *The Importance of Initial Conditions in Seasonal Predictions of Antarctic Sea Ice**

**1  Overall evaluation:**

This study employs two versions of the Australian Bureau of Meteorology's ACCESS seasonal forecasting system (ACCESS-S1 and ACCESS-S2), which share an identical model configuration but differ in their initialization strategies, to investigate the role of initial conditions in Antarctic sea ice prediction. The results are promising and broadly consistent with findings from previous studies. Overall, the manuscript is suitable for publication in The Cryosphere. However, I recommend that the authors address the comments outlined below before publication.

**2  General comments:**

**Improper methodology for addressing the stated objective**:
The manuscript emphasizes the importance of sea ice initialization (e.g., L28–29, L56–57, and L85). While ACCESS-S1 and ACCESS-S2 share identical model configurations (e.g., L54 and L65–73) and both use atmospheric initial conditions derived from ERA-Interim (L74), their ocean and sea ice initial conditions differ substantially in both assimilation methods and observational datasets (L74–84). Consequently, a direct comparison between ACCESS-S1 and ACCESS-S2 does not isolate the impact of sea ice initialization alone, as ocean initialization is known to play a critical role in sea ice predictability. Furthermore, the ensemble generation strategies differ between the two systems (L93–100), which may also affect the comparison of ensemble-mean results.

I therefore suggest revising the stated objective of the manuscript and explicitly analyzing the combined impacts of ocean, atmosphere, and sea ice initialization, rather than attributing the results solely to sea ice initialization, given the fully coupled nature of the system.

**Quality of figures**:

I have several specific comments regarding the figures. Improving their quality and clarity would substantially enhance the overall quality of the manuscript and improve its readability.

**3   Specific comments:**

L64, Section 2.1 (ACCESS-S1 and ACCESS-S2): Please ensure consistency between the section title and its content. Clarify whether this section sufficiently describes the key differences between ACCESS-S1 and ACCESS-S2 that are relevant to this study.

L125, Section 2.3: The current version of Section 2.3 primarily describes RMSE and skill scores. However, bias is extensively used in Section 3 (Results) and should therefore be formally introduced and documented in the methodology section. In addition, correlation is also employed (e.g., Fig. 5) and is a widely used metric for evaluating seasonal sea ice forecasts; it should be described as well. Furthermore, the authors should report the statistical significance or confidence intervals for RMSE and correlation.

L126–127: It is unclear how the ensemble mean sea ice extent (SIE) is calculated. Do the authors first compute the ensemble mean of sea ice concentration (SIC) and then derive SIE from the mean SIC, or do they compute SIE separately for each ensemble member and then analyze the ensemble mean of SIE? This should be explicitly clarified in the text.

Figures 3 and 4: These figures are difficult to read. Adding latitude–longitude grid lines would improve readability. In addition, narrowing the colorbar range (e.g., from $-1$ to 1 to $-0.5$ to 0.5) may help highlight relevant spatial patterns.

L194: The term "skillful" is ambiguous here. Please clarify whether it refers to skill assessed using RMSE, correlation, or another metric.

L214–215: It appears that the magnitude of sea ice anomalies in ACCESS-S2 is too small, rather than too large, as currently stated. Please verify and revise this interpretation.

L245: "June cannot be forecasted..."?

Figures 9 and 10: Figure 10 largely overlaps with the information presented in Figure 9. It may not be necessary to include Figure 10, or its added value should be better justified.

Figures 11–13: These figures are difficult to interpret. Consider using an alternative colorbar, narrowing the colorbar range, and adding longitude grid lines to improve clarity.

Caption of Figure 14: Please clarify the lead time shown (e.g., 1, 3, and 30 days?).

L277–278: As shown in Fig. 14c1, the two systems clearly differ.

L279–280: How is the standard deviation computed? Is it calculated across years, ensemble members, or both? Please clarify.

Figure 14: The methodology used to decompose the forecast error spread into the mean standard deviation and the standard deviation of the ensemble-mean error is not sufficiently clear and should be explained in more detail.

L307–308: Please clarify how ACCESS-S1 updates sea ice states during the assimilation step. Specifically, does the assimilation update both sea ice concentration and volume across different thickness bins, or does it retain the prior thickness distribution and adjust sea ice volume proportionally during the post-processing of the assimilation step?

L313–314: I kindly disagree with the authors' statement here; please refer to my general comments for further explanation.

Additional references: I recommend citing the following two studies, which are highly relevant to the discussion of Antarctic sea ice initialization and predictability:
https://journals.ametsoc.org/view/journals/clim/34/15/JCLI-D-20-0965.1.xml
https://agupubs.onlinelibrary.wiley.com/doi/full/10.1029/2024MS004382